# Copper and Zinc Sulfates Suppress *Streptomyces* spp. and Enhance Potato Resistance via Thaxtomin A Inhibition and Defense Gene Regulation

**DOI:** 10.3390/microorganisms13061288

**Published:** 2025-05-31

**Authors:** Nianzhou Chen, Shuning Zhou, Shuo Yan, Xin Yuan, Weiqi Jiao, Xinbo Wang, Jie Liu, Xuanzhe Zhang

**Affiliations:** Department of Plant Pathology, College of Plant Protection, Northeast Agricultural University, Harbin 150038, China; c841170003@gmail.com (N.C.); zhousn2001@163.com (S.Z.); shuoyan666777888@163.com (S.Y.); yxin7996@gmail.com (X.Y.); qaujwq@163.com (W.J.); 18800469101@163.com (X.W.); 2724607170lj@gmail.com (J.L.)

**Keywords:** *Streptomyces* spp., Thaxtomin A, potato resistance, defense genes, copper sulfate, zinc sulfate

## Abstract

Potato (*Solanum tuberosum* L.) is a major staple crop globally, yet its production is severely impacted by common scab, a disease caused by *Streptomyces* spp., leading to substantial economic losses. This study evaluated copper sulfate (CuSO_4_) and zinc sulfate (ZnSO_4_) as potential control agents for common scab, focusing on their antimicrobial properties and effects on potato resistance mechanisms. Both CuSO_4_ and ZnSO_4_ exhibited dose-dependent inhibition of *Streptomyces* spp., significantly reducing the production of the pathogenic toxin Thaxtomin A by 57.02% and 41.29%, respectively. Electrical conductivity assays indicated their disruptive effects on cell membrane integrity, and HPLC confirmed their suppression of toxin production. Pot experiments showed that these treatments enhanced plant growth, chlorophyll content, and defense enzyme activities (SOD, POD, CAT, PPO), while reducing malondialdehyde (MDA) levels. qPCR analysis revealed upregulation of defense-related genes (PR1, PR3, PR9, SOD1, HSF1). Field trials demonstrated disease control efficiencies of 56.58% and 59.06% for CuSO_4_ and ZnSO_4_, respectively, with ZnSO_4_ increasing yield by 19.29%. These findings highlight CuSO_4_ and ZnSO_4_ as effective agents for suppressing *Streptomyces* spp. and enhancing potato resistance, offering practical value for sustainable potato production systems.

## 1. Introduction

Potato (*Solanum tuberosum* L.) ranks as the fourth most important staple crop worldwide, contributing significantly to global food security due to its high nutritional value and adaptability to diverse agroecological conditions [1,2]. Despite its agricultural importance, potato production faces substantial challenges from diseases, notably potato common scab (PCS), caused by several pathogenic *Streptomyces* species, including *S. scabies*, *S. acidiscabies*, and *S. turgidiscabies* [3,4]. PCS manifests as corky, pitted lesions on tuber surfaces, severely compromising their aesthetic quality and market value, leading to economic losses estimated at millions annually in major potato-producing regions [5]. The disease’s widespread prevalence, cryptic nature, and persistence in soil make it a formidable challenge for growers, necessitating innovative and sustainable management strategies.

The pathogenicity of PCS is primarily driven by the phytotoxin Thaxtomin A, produced by pathogenic *Streptomyces* spp., which disrupts plant cell wall integrity, induces necrosis, and facilitates disease symptom development [6,7]. Beyond potatoes, Thaxtomin A’s phytotoxicity affects other crops, such as sugar beets and radishes, amplifying its agricultural impact [8]. Current PCS management strategies encompass breeding resistant cultivars, cultural practices, chemical treatments, and biological control, yet each approach has significant limitations. Developing resistant cultivars is a protracted and costly improvised breed of potatoes is time-consuming and often fails to provide complete immunity due to the genetic variability of *Streptomyces* spp. [9]. Cultural practices, such as crop rotation and soil pH adjustment, offer limited efficacy and are difficult to implement consistently across diverse farming systems [10]. Chemical controls, including fumigants and antibiotics, risk environmental contamination and the emergence of resistant pathogen strains, while biological controls, such as antagonistic microorganisms, are slow-acting and context-dependent [11,12]. These challenges underscore the urgent need for safe, effective, and environmentally sustainable alternatives to manage PCS.

Copper (Cu) and zinc (Zn) are essential micronutrients with well-documented antimicrobial properties, offering a promising avenue for integrated disease management. Copper- and zinc-based compounds have been shown to inhibit a broad spectrum of plant pathogens by disrupting cellular structures, suppressing toxin production, and enhancing host resistance [13,14]. For instance, copper ions can destabilize bacterial membranes through oxidative stress, while zinc modulates enzymatic activities critical to pathogen virulence [15,16]. Recent studies have highlighted their potential in reducing Thaxtomin A synthesis and mitigating PCS severity in controlled settings [17]. However, their precise mechanisms in *Streptomyces* spp. suppression and plant resistance induction remain underexplored. This study investigated the efficacy of copper sulfate (CuSO_4_) and zinc sulfate (ZnSO_4_) in controlling PCS through a multifaceted approach, including in vitro antimicrobial assays, Thaxtomin A quantification, pot experiments, and molecular analyses of defense gene expression. By elucidating their roles in pathogen suppression and potato resistance enhancement, this research aims to provide actionable insights for developing novel, sustainable strategies to combat PCS and support global potato production [18,19].

## 2. Materials and Methods

### 2.1. Materials and Culture Media

The *Streptomyces* strains used in this study included HRB-1 (*S. scabies*), HL-12 (*S. rochei*), JX-15 (*S. lavendulae*), JMS-3 (*S. acidiscabies*), and AC-15 (*S. bottropensis*), which were isolated from potato-common-scab-infected soil and tuber samples using Gause’s No.1 medium, identified through 16S rRNA gene sequencing and morphological characterization, and preserved in 20% glycerol at −80 °C by our research team [4,20]. The test plant was the potato cultivar “Yujing 885”. The primary reagents included copper sulfate (CuSO_4_), zinc sulfate (ZnSO_4_), and widely used agricultural antibiotics (e.g., streptomycin sulfate and Zhongshengmycin), ensuring comparability of results. Other chemical reagents such as propidium iodide (PI) stain, Thaxtomin A, and chromatographic-grade acetonitrile were purchased from reputable suppliers and met analytical grade specifications. The culture media included ISP2, modified Gause’s No.1, oat bran liquid medium, and 1% water agar, all of which were sterilized at 120 °C for 20 min before use.

### 2.2. Antibacterial Effects of Copper and Zinc

The antibacterial activity of CuSO_4_ and ZnSO_4_ against *Streptomyces* spp. was evaluated using the disk diffusion method. The CuSO_4_ concentrations (0.04–0.16 g/mL) and ZnSO_4_ concentrations (0.10–0.50 g/mL) were selected based on preliminary experiments and relevant literature to ensure optimal pathogen suppression without phytotoxicity [13,14]. A 100 μL aliquot of bacterial suspension was evenly spread onto ISP2 agar plates, and sterilized filter paper discs (0.6 cm in diameter) were placed at the center, each loaded with 15 μL of the corresponding concentrations of the reagents. Three biological replicates were performed for each concentration of CuSO_4_ and ZnSO_4_ to ensure reproducibility of the results. After incubation at 28 °C for 7 days, inhibition zone diameters were measured, and inhibition rates were calculated. Additionally, the inhibitory effects of agricultural streptomycin sulfate, Zhongshengmycin, Polyoxin, Kasugamycin, and Chunlei·Wangtong at gradient concentrations were tested. Half-maximal inhibitory concentrations (EC_50_) were calculated using regression analysis to evaluate the toxicity of the reagents. Data were analyzed using SPSS 25.0 with ANOVA and Tukey’s test (*p* < 0.05).

### 2.3. Mechanisms of Copper and Zinc Effects on Streptomyces Scabies

The impact of CuSO_4_ and ZnSO_4_ on bacterial cell membrane permeability and nuclear integrity was assessed using electrical conductivity and PI staining assays. HRB-1 suspensions (100 μL) were inoculated into ISP2 liquid media containing different concentrations of CuSO_4_ (0.001, 0.0015, 0.002 g/mL) and ZnSO_4_ (0.002, 0.0035, 0.005 g/mL) and cultured in a shaker. Changes in electrical conductivity of the fermentation broth were measured at 0, 1, 2, 3, and 4 h to evaluate cell membrane damage. Bacterial cells were collected by centrifugation, stained with PI, and observed under a fluorescence microscope to assess nuclear damage. Additionally, high-performance liquid chromatography (HPLC) was used to analyze the Thaxtomin A production by *S. scabies* under different treatments.

### 2.4. Effects of Copper and Zinc on Potato Growth

Healthy, pre-sprouted potato tubers were cut into seed pieces and planted in pots containing equal amounts of soil and compound fertilizer, with a planting depth of 10 cm. After one month, roots were inoculated with a mixed suspension of *Streptomyces* spp. at a concentration of 1 × 10^8^ cfu/mL, applied every 7 days for three consecutive treatments. Subsequently, different concentrations of CuSO_4_ (0.001, 0.0015, 0.002 g/mL) and ZnSO_4_ (0.002, 0.0035, 0.005 g/mL) were applied, while the control group received sterile water, and the positive control group was treated with 0.00035 g/mL agricultural streptomycin sulfate. At the end of the experiment, plant growth parameters, including height, stem diameter, fresh weight, and dry weight, were measured. Relative chlorophyll content was determined using a chlorophyll meter to evaluate the growth-promoting effects of copper and zinc treatments.

### 2.5. Analysis of Resistance Enzyme Activity and Gene Expression

After treatment with CuSO_4_ (0.002 g/mL) and ZnSO_4_ (0.005 g/mL), plant samples were collected at 4, 8, 12, 36, and 60 h to measure the activities of resistance-related enzymes, including superoxide dismutase (SOD), peroxidase (POD), catalase (CAT), polyphenol oxidase (PPO), and malondialdehyde (MDA). Total RNA was extracted from samples ground in liquid nitrogen and reverse-transcribed into cDNA. Quantitative real-time PCR (qPCR) was performed to analyze the expression of resistance-related genes (PR1, PR3, PR9, SOD1, HSF1). Gene expression levels were calculated using the 2^−ΔΔCt^ method.

### 2.6. Soil Safety Assessment

To evaluate the environmental safety of CuSO_4_ and ZnSO_4_ applications, soil samples were collected from the pot experiments described in Section 2.4 after the final treatment application. For each treatment (CK, CuSO_4_ at 0.001 g/mL, 0.0015 g/mL, and 0.002 g/mL, and ZnSO_4_ at 0.002 g/mL, 0.0035 g/mL, and 0.005 g/mL), three replicate soil samples were taken from the top 0–15 cm layer, air-dried, and sieved through a 2 mm mesh. The copper (Cu) and zinc (Zn) contents in the soil were determined using atomic absorption spectrophotometry (AAS, PerkinElmer Analyst 800,PerkinElmer, Inc., Waltham, MA, USA) following the Chinese standard method (HJ 491-2019) [21]. Briefly, 0.5 g of each soil sample was digested with a mixture of nitric acid (HNO_3_) and hydrochloric acid (HCl) (3:1 ratio) in a microwave digestion system, and the resulting solution was analyzed for Cu and Zn concentrations. Calibration curves were prepared using standard solutions of Cu and Zn, and all measurements were performed in triplicate to ensure accuracy. The results were compared against the risk screening values for agricultural land as per the Chinese Soil Environmental Quality Standard (GB15618-2018) [22], which sets limits at 100 mg/kg for Cu and 300 mg/kg for Zn.

### 2.7. Statistical Analysis

All data were analyzed using SPSS 25.0 with one-way ANOVA followed by Tukey’s post hoc test (*p* < 0.05), unless specified. Analyses aimed to evaluate treatment and time effects on biological responses. Antibacterial effects (Appendix A) compared inhibition zones across concentrations within each Streptomyces strain. Thaxtomin A levels (Appendix A) were compared across treatments. Growth parameters (Appendix A) were compared across treatments. Enzyme activities and MDA levels were compared among treatments and time points (4, 8, 12, 36, 60 h). Gene expression was compared among time points within treatments and among treatments (CK, 0.002 g/mL CuSO_4_, 0.005 g/mL ZnSO_4_) at each time point. Soil Cu and Zn contents (Appendix A) were compared across concentrations within treatments. EC_50_ values (Table 1) were calculated via regression analysis. Data are mean ± SE from three replicates. Detailed results are available from the corresponding author upon request.

**Table 1 microorganisms-13-01288-t001:** Determination of the in vitro virulence of copper sulfate and zinc sulfate against five *Streptomyces* spp.

Medicament	Strains	Regression Equation	R^2^	EC_50_ (g/mL)
CuSO_4_	HRB-1 (*S. scabies*)	Y = 0.7256x + 5.2297	0.9709	0.4824
	HL-12 (*S. rochei*)	Y = 1.8788x + 5.7666	0.9708	0.3908
	JX-15 (*S. lavendulae*)	Y = 1.4410x + 6.1002	0.9841	0.1724
	JMS-3 (*S. acidiscabies*)	Y = 2.0650x + 6.3676	0.9997	0.2176
	AC-15 (*S. bottropensis*)	Y = 0.7739x + 5.2670	0.9426	0.4518
ZnSO_4_	HRB-1 (*S. scabies*)	Y = 0.5828x + 5.0079	0.9872	0.9693
	HL-12 (*S. rochei*)	Y = 1.2072x + 5.0065	0.9988	0.9877
	JX-15 (*S. lavendulae*)	Y = 0.8400x + 4.9510	0.9775	1.1438
	JMS-3 (*S. acidiscabies*)	Y = 1.0692x + 4.8013	0.8255	1.5341
	AC-15 (*S. bottropensis*)	Y = 0.9122x + 4.8199	0.9917	1.5756

Note: Regression equations, R^2^ values, and EC_50_ values were calculated using inhibition zone diameters (cm) presented in Figure 1a. The regression analysis quantifies the dose–response relationship between CuSO_4_ or ZnSO_4_ concentrations (g/mL) and inhibition zone diameters for each *Streptomyces* strain.

**Figure 1 microorganisms-13-01288-f001:**
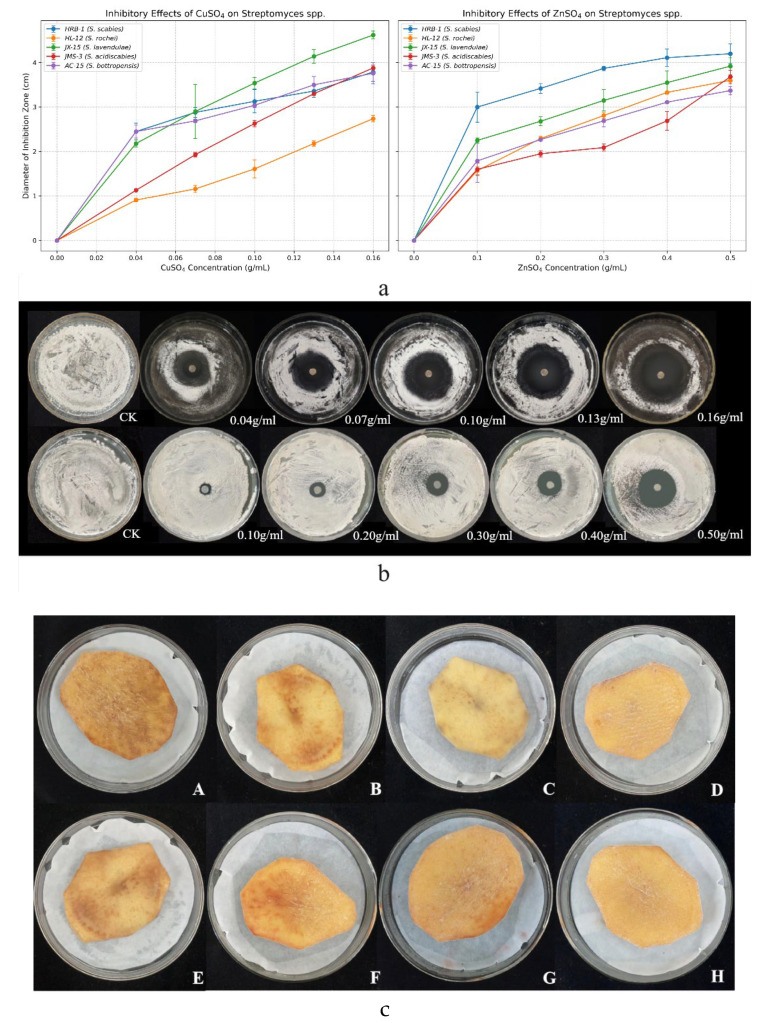
(a) Inhibitory effects of copper sulfate and zinc sulfate on five *Streptomyces* spp. at varying concentrations. (b) Specific inhibitory effects on selected strains. (first row) Inhibitory effects of copper sulfate on strain AC-15 at different concentrations. (second row) Inhibitory effects of zinc sulfate on strain JMS-3 at different concentrations. (c) Infection effects of copper sulfate (CuSO_4_) and zinc sulfate (ZnSO_4_) on Streptomyces scabies in potato chips in vitro. Note: A, E: CK; B: 0.001 g/mL CuSO_4_; C: 0.0015 g/mL CuSO_4_; D: 0.002 g/mL CuSO_4_; F: 0.002 g/mL ZnSO_4_; G: 0.0035 g/mL ZnSO_4_; H: 0.005 g/mL ZnSO_4_.

## 3. Results

### 3.1. Antibacterial Effects of Copper and Zinc on Streptomyces *spp.* Causing Potato Common Scab

The results showed that copper sulfate (CuSO_4_) and zinc sulfate (ZnSO_4_) exhibited significant antibacterial activity against five *Streptomyces* strains causing potato common scab, with their inhibitory effects increasing in a dose-dependent manner (Figure 1a). At a concentration of 0.1 g/mL CuSO_4_, the inhibition zone diameters for strains HRB-1 (*S. scabies*) and AC-15 (*S. bottropensis*) were 3.13 ± 0.26 cm and 3.04 ± 0.05 cm, respectively, corresponding to inhibition rates of 22.02% and 19.84% (Figure 1a). ZnSO_4_ at 0.5 g/mL showed a maximum inhibition zone diameter of 4.20 ± 0.22 cm against strain HRB-1 (*S. scabies*), corresponding to a 42.86% inhibition rate, outperforming other strains (Figure 1a). One-way ANOVA followed by Tukey’s post hoc test (*p* < 0.05) confirmed significant differences among concentrations for each strain, with detailed results presented in Table 1. Regression analysis revealed a strong dose–response relationship, with R^2^ values ranging from 0.9708 to 0.9997 for CuSO_4_ and 0.8255 to 0.9988 for ZnSO_4_ (Table 1). The EC_50_ values for CuSO_4_ ranged from 0.1724 g/mL (JX-15) to 0.4824 g/mL (HRB-1), while for ZnSO_4_, they ranged from 0.9693 g/mL (HRB-1) to 1.5756 g/mL (AC-15), indicating varying sensitivities among strains (Table 1). Detached tuber assays further revealed that both CuSO_4_ and ZnSO_4_ effectively inhibited the growth of *Streptomyces* spp., as evidenced by the inhibition zones of strains AC-15 and JMS-3 at different concentrations (Figure 1). These findings confirm that CuSO_4_ and ZnSO_4_ are safe for use on potato tubers at the tested concentrations.

Toxicity assays conducted in vitro indicated that CuSO_4_ was most effective against strain JX-15, with an EC_50_ value of 0.1724 g/mL, while ZnSO_4_ demonstrated the highest sensitivity to strain HRB-1, with an EC_50_ value of 0.9693 g/mL (Table 1). Detached tuber assays revealed that both CuSO_4_ and ZnSO_4_ significantly inhibited browning of potato slices induced by *S. scabies* (Figure 1b). Potato slices treated with CuSO_4_ or ZnSO_4_ maintained their shape and showed no signs of decay, whereas those in the untreated control group exhibited progressive browning over time. These findings confirm that CuSO_4_ and ZnSO_4_ are safe for use on potato tubers at the tested concentrations.

### 3.2. Disruptive Effects of Copper and Zinc on the Cellular Structure of Pathogens

The mechanisms of action of copper and zinc were further elucidated through electrical conductivity and propidium iodide (PI) staining assays. After treatment with copper sulfate (CuSO_4_, 0.002 g/mL), the electrical conductivity of the solution increased to 201.39 µS/cm, while zinc sulfate (ZnSO_4_, 0.005 g/mL) treatment resulted in a conductivity increase to 297.05 µS/cm, indicating that high concentrations of these agents significantly disrupted the permeability of the bacterial cell membrane (Figure 2).

Moreover, PI staining assays revealed intense red fluorescence signals in the nuclei of bacteria treated with CuSO_4_ or ZnSO_4_, indicating nucleic acid leakage. These findings further confirm the damaging effects of CuSO_4_ and ZnSO_4_ on the bacterial cell nucleus (Figure 3).

### 3.3. Inhibitory Effects of Copper and Zinc on the Production of the Virulence Toxin Thaxtomin A

High-performance liquid chromatography (HPLC) analysis demonstrated that copper sulfate (CuSO_4_) and zinc sulfate (ZnSO_4_) significantly suppressed the production of the virulence toxin Thaxtomin A by *Streptomyces* spp. (Figure 4a). In the control group, the toxin content was 487.75 mg/L. CuSO_4_ treatments (0.001–0.002 g/mL) progressively reduced toxin levels, achieving a maximum inhibition rate of 57.02%. ZnSO_4_ showed comparable effects, indicating its potential for practical scab management. Similarly, treatment with 0.005 g/mL ZnSO_4_ decreased the toxin content to 286.36 mg/L (inhibition rate of 41.29%) (Figure 4b). These results indicate that CuSO_4_ and ZnSO_4_ significantly reduce the pathogenicity of *Streptomyces* spp. by inhibiting the production of Thaxtomin A.

### 3.4. Effects of Copper and Zinc on Potato Growth and Disease Resistance

Treatments with copper sulfate (CuSO_4_) and zinc sulfate (ZnSO_4_) significantly enhanced potato plant growth and development (Appendix A). After treatment with 0.005 g/mL ZnSO_4_, plant height increased to 43.48 ± 3.22 cm compared to 40.24 ± 1.36 cm in the control group (CK), with fresh weight and dry weight rising to 126.14 ± 9.74 g and 42.25 ± 3.94 g, respectively (*p* < 0.05, Appendix A). Treatment with 0.002 g/mL CuSO_4_ increased stem diameter to 1.13 ± 0.03 cm, significantly higher than 0.98 ± 0.07 cm in the CK (*p* < 0.05, Appendix A). These improvements are visualized in Figure 5, with detailed statistical results presented in Appendix A.

In addition, both CuSO_4_ and ZnSO_4_ treatments significantly enhanced leaf chlorophyll content, measured as 42.51 SPAD and 42.81 SPAD, respectively, compared to 40.20 SPAD in the control group (CK) (Figure 6). Antioxidant enzyme activity analysis showed that treatments with 0.002 g/mL CuSO_4_ and 0.005 g/mL ZnSO_4_ significantly increased the activities of SOD, POD, CAT, and PPO, while reducing MDA levels compared to the control (CK) (Figure 7). Significant differences among treatments were observed at most time points (4, 8, 12, 36, and 60 h) (*p* < 0.05, one-way ANOVA followed by Tukey’s post hoc test), as indicated by lowercase letters in Figure 7. For example, at 12 h, SOD activity under CuSO_4_ treatment reached 140 ± 3.5 U/g, significantly higher than 90 ± 2.2 U/g in CK, while MDA levels at 36 h decreased to 50 ± 1.5 nmol/g under ZnSO_4_ treatment compared to 70 ± 2.0 nmol/g in CK. These findings indicate that CuSO_4_ (0.002 g/mL) and ZnSO_4_ (0.005 g/mL) play a critical role in mitigating oxidative damage and promoting disease resistance in potato plants by enhancing their physiological and biochemical defenses, suggesting their potential application in sustainable agricultural practices.

### 3.5. Dynamic Changes in the Expression of Disease-Resistance Genes

Quantitative real-time PCR (qPCR) analysis revealed that treatments with 0.002 g/mL CuSO_4_ and 0.005 g/mL ZnSO_4_ significantly upregulated the expression of key disease-resistance genes, including PR1, PR3, PR9, SOD1, and HSF1, compared to the control (CK) (Figure 8). Expression levels were calculated using the 2^−ΔΔCt^ method, with data representing mean fold changes ± standard error (SE) from three biological replicates with three technical replicates each. One-way ANOVA followed by Tukey’s post hoc test (*p* < 0.05) confirmed significant differences both among time points (4, 8, 12, 36, and 60 h) within each treatment and among treatments at each time point. For instance, the expression level of PR3 peaked at 48 h after ZnSO_4_ treatment, reaching 5.09 ± 0.15-fold relative to CK (1.00 ± 0.05-fold, *p* < 0.05), while CuSO_4_ treatment at the same time point showed a 4.20 ± 0.12-fold increase (*p* < 0.05). Similar trends were observed for PR1, PR9, SOD1, and HSF1, with significant upregulation at 8–48 h under both treatments. Complete statistical results for all genes and time points are available from the corresponding author upon reasonable request.

These results demonstrate that CuSO_4_ and ZnSO_4_ enhance plant disease resistance through a multifaceted mechanism. This involves the upregulation of defense-related genes, activation of systemic acquired resistance (SAR), and strengthening of antioxidative defense systems. The increased expression of SOD1 highlights a role in mitigating oxidative stress, while the elevated HSF1 levels suggest enhanced stress signaling. Together, these responses synergistically improve the plant’s ability to counteract biotic and abiotic stressors.

Our findings provide a molecular basis for CuSO_4_ and ZnSO_4_-induced resistance, supporting their potential as effective agrochemical agents for crop protection. Further research should focus on elucidating downstream regulatory networks and assessing the field applicability of these treatments across diverse environmental conditions.

### 3.6. Soil Safety Assessment

Soil copper (Cu) and zinc (Zn) contents were measured after treatments as described in Section 2.6. For CuSO_4_ treatments, soil Cu content increased from 23.04 ± 0.43 mg/kg in the control (CK) to 56.90 ± 1.44 mg/kg at 0.001 g/mL, 66.26 ± 0.98 mg/kg at 0.0015 g/mL, and 74.09 ± 1.13 mg/kg at 0.002 g/mL (Appendix A). For ZnSO_4_ treatments, soil Zn content rose from 15.44 ± 1.00 mg/kg in CK to 34.72 ± 1.02 mg/kg at 0.002 g/mL, 66.55 ± 0.95 mg/kg at 0.0035 g/mL, and 94.67 ± 0.95 mg/kg at 0.005 g/mL (Appendix A). All measured Cu and Zn levels remained below the risk screening values for agricultural land (100 mg/kg for Cu and 300 mg/kg for Zn) as per the Chinese Soil Environmental Quality Standard (GB15618-2018), indicating that the applied concentrations are safe for short-term agricultural use.

## 4. Discussion

This study provides a comprehensive evaluation of copper sulfate (CuSO_4_) and zinc sulfate (ZnSO_4_) as sustainable agents for managing potato common scab (PCS), caused by *Streptomyces* spp., through pathogen suppression, toxin inhibition, and enhancement of plant resistance. By integrating in vitro assays, pot experiments, field trials, and molecular analyses, our results highlight the multifaceted benefits of these micronutrients in improving potato health and yield while addressing environmental safety concerns.

CuSO_4_ and ZnSO_4_ exhibited potent dose-dependent antibacterial activity against five *Streptomyces* strains, with inhibition zone diameters reaching 3.79 cm at 0.002 g/mL CuSO_4_ and 4.20 cm at 0.005 g/mL ZnSO_4_, compared to 0.00 cm in the control (Figure 1a). These translated into inhibition rates of 22.02% and 42.86%, respectively, surpassing the efficacy of streptomycin in some strains (Figure 1a). Detached tuber assays further confirmed their effectiveness, reducing lesion expansion and browning in potato slices (Figure 1b), indicating safety for tuber application at tested concentrations. Mechanistically, electrical conductivity assays showed that 0.002 g/mL CuSO_4_ and 0.005 g/mL ZnSO_4_ increased bacterial membrane permeability to 201.39 µS/cm and 297.05 µS/cm, respectively (Figure 2), while PI staining revealed nucleic acid leakage (Figure 3). These findings suggest that Cu^2+^ and Zn^2+^ ions induce oxidative stress, disrupting bacterial cell structures, consistent with Quaglia et al. (2021), who reported similar membrane-destabilizing effects on *Pseudomonas syringae* [13]. Compared to streptomycin, which risks resistance development [12], CuSO_4_ and ZnSO_4_ offer a more sustainable approach due to their natural occurrence and lower resistance potential, making them promising alternatives for PCS management.

A critical aspect of PCS pathogenicity is the production of Thaxtomin A, which disrupts plant cell wall integrity and induces necrosis [7]. CuSO_4_ and ZnSO_4_ significantly inhibited Thaxtomin A production, achieving maximum inhibition rates of 57.02% (0.002 g/mL CuSO_4_) and 41.29% (0.005 g/mL ZnSO_4_), corresponding to reductions of 278.10 mg/L and 201.39 mg/L, respectively (Figure 4b). This aligns with Hassan et al. (2021), who noted metal ions’ role in toxin suppression [17], but our study extends these findings by linking Thaxtomin A reduction to enhanced plant resistance. The decrease in Thaxtomin A likely reduces disease severity by limiting pathogen virulence, creating a favorable environment for subsequent plant defense responses.

At the plant level, CuSO_4_ and ZnSO_4_ significantly enhanced growth parameters and chlorophyll content, supporting their dual role in disease resistance and growth promotion (Figure 5 and Figure 6). Specifically, 0.005 g/mL ZnSO_4_ significantly increased plant height to 43.48 ± 3.22 cm (vs. 40.24 ± 1.36 cm in CK), fresh weight to 126.14 ± 9.74 g, and dry weight to 42.25 ± 3.94 g, while 0.002 g/mL CuSO_4_ significantly raised stem diameter to 1.13 ± 0.03 cm (vs. 0.98 ± 0.07 cm in CK) and chlorophyll content to 42.51 ± 0.80 SPAD (vs. 40.20 ± 0.75 SPAD in CK) (*p* < 0.05, Appendix A). These improvements are attributable to Cu and Zn’s roles as essential micronutrients in chlorophyll synthesis and enzymatic functions [23]. The heatmap analysis (Figure 5) showed consistent enhancements across multiple growth indicators, with normalized values reaching up to 1.0, indicating maximum improvement relative to CK. Similar growth-promoting effects of Zn have been reported in maize and rice [24], but our study uniquely demonstrates these benefits in the context of PCS management, linking growth enhancement to improved disease resistance.

The treatments also bolstered potato resistance by modulating oxidative stress and defense gene expression. CuSO_4_ and ZnSO_4_ significantly increased the activities of SOD, POD, CAT, and PPO, while reducing MDA levels (Figure 7), indicating a reduction in lipid peroxidation and oxidative damage caused by pathogen-induced ROS. This aligns with Kopecky et al. (2021), who found Zn supplementation enhanced SOD and CAT activities under biotic stress [14]. qPCR analysis revealed significant upregulation of defense-related genes (PR1, PR3, PR9, SOD1, HSF1), with PR3 expression peaking at 5.09-fold after ZnSO_4_ treatment (Figure 8). PR3, encoding a chitinase, likely degrades pathogen cell walls, while SOD1 mitigates oxidative stress, and HSF1 enhances stress signaling [25,26]. The interplay between Thaxtomin A reduction, antioxidant enzyme activation, and defense gene upregulation forms a synergistic defense mechanism, reducing disease severity while promoting plant vigor. This integrated response highlights the potential of CuSO_4_ and ZnSO_4_ to enhance systemic acquired resistance (SAR), a mechanism previously noted in other crops under stress [26].

Field trials validated these findings, showing control efficiencies of 56.58% for CuSO_4_ and 59.06% for ZnSO_4_, with ZnSO_4_ increasing yield by 19.29%. These efficiencies are comparable to or exceed those of chemical controls like streptomycin [19], but with added sustainability benefits. However, regular use of CuSO_4_ and ZnSO_4_ raises concerns about potential soil contamination with heavy metals (Cu and Zn). Soil safety assessments indicated that at the tested concentrations (up to 0.002 g/mL CuSO_4_ and 0.005 g/mL ZnSO_4_), soil Cu and Zn levels remained below the risk screening values of the Chinese Soil Environmental Quality Standard (GB15618-2018), at 74.09 mg/kg for Cu (limit: 100 mg/kg) and 94.67 mg/kg for Zn (limit: 300 mg/kg) (Appendix A). While these levels are safe for agricultural use in the short term, repeated applications over time could lead to accumulation, potentially exceeding safe thresholds and affecting soil microbial communities and long-term soil health [13]. For instance, excessive Cu accumulation has been linked to reduced microbial diversity in agricultural soils, while high Zn levels may impact soil pH and nutrient availability [18]. To mitigate these risks, we recommend regular monitoring of soil Cu and Zn levels and exploring integrated management strategies, such as combining CuSO_4_ and ZnSO_4_ with biological controls to reduce application frequency.

This study provides actionable insights for PCS management, with CuSO_4_ and ZnSO_4_ offering a sustainable alternative to chemical controls. We recommend field application at 0.002 g/mL CuSO_4_ and 0.005 g/mL ZnSO_4_, applied as foliar sprays or soil drenches, to balance efficacy and safety. Future research should focus on optimizing application strategies, evaluating synergistic effects with biological controls, and assessing long-term impacts on soil health and microbial ecology [27]. Additionally, molecular studies targeting the downstream regulatory networks of PR3 and SOD1 could further elucidate the pathways through which CuSO_4_ and ZnSO_4_ enhance resistance, paving the way for precision agriculture applications.

In conclusion, CuSO_4_ and ZnSO_4_ offer a sustainable strategy for PCS management by suppressing *Streptomyces* spp., reducing Thaxtomin A production, and enhancing potato growth and resistance through physiological and molecular mechanisms. While their short-term soil safety is confirmed, careful management is required to prevent long-term heavy metal accumulation, ensuring their viability in sustainable potato production.

## Figures and Tables

**Figure 2 microorganisms-13-01288-f002:**
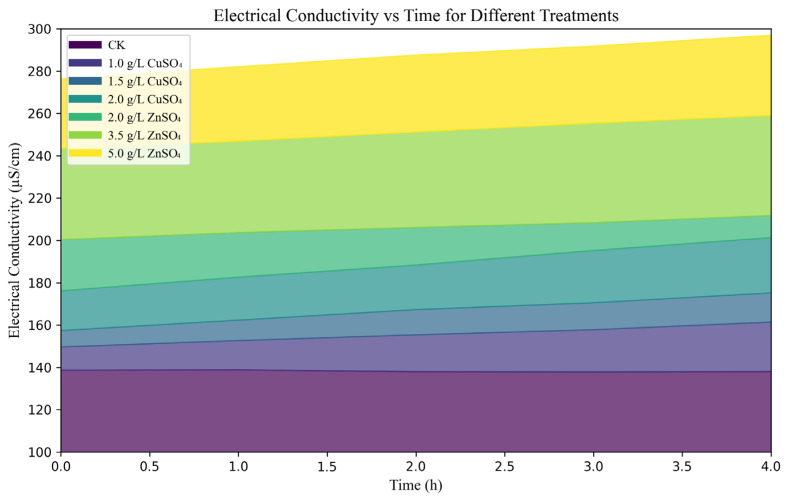
Effects of copper sulfate (CuSO_4_) and zinc sulfate (ZnSO_4_) on the cell membrane conductivity of *Streptomyces scabies*.

**Figure 3 microorganisms-13-01288-f003:**
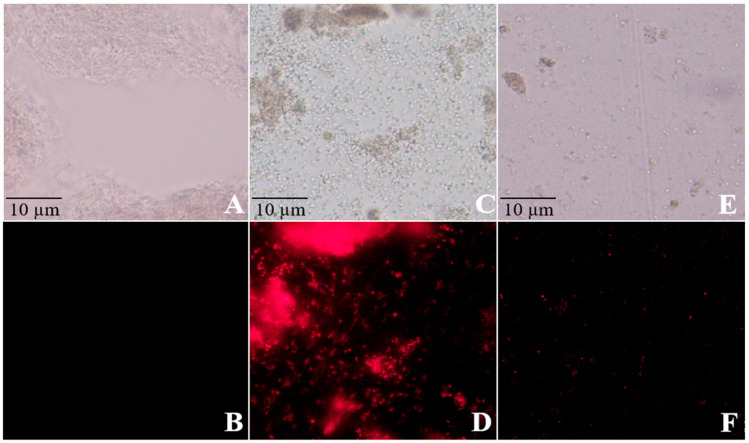
Effects of copper sulfate (CuSO_4_) and zinc sulfate (ZnSO_4_) on the cell nucleus of *Streptomyces scabies*. Note: (A): CK had no fluorescence, (B): CK had fluorescence, (C): CuSO_4_ treatment had no fluorescence, (D): CuSO_4_ treatment had fluorescence, (E): ZnSO_4_ treatment had no fluorescence, (F): ZnSO_4_ treatment had fluorescence.

**Figure 4 microorganisms-13-01288-f004:**
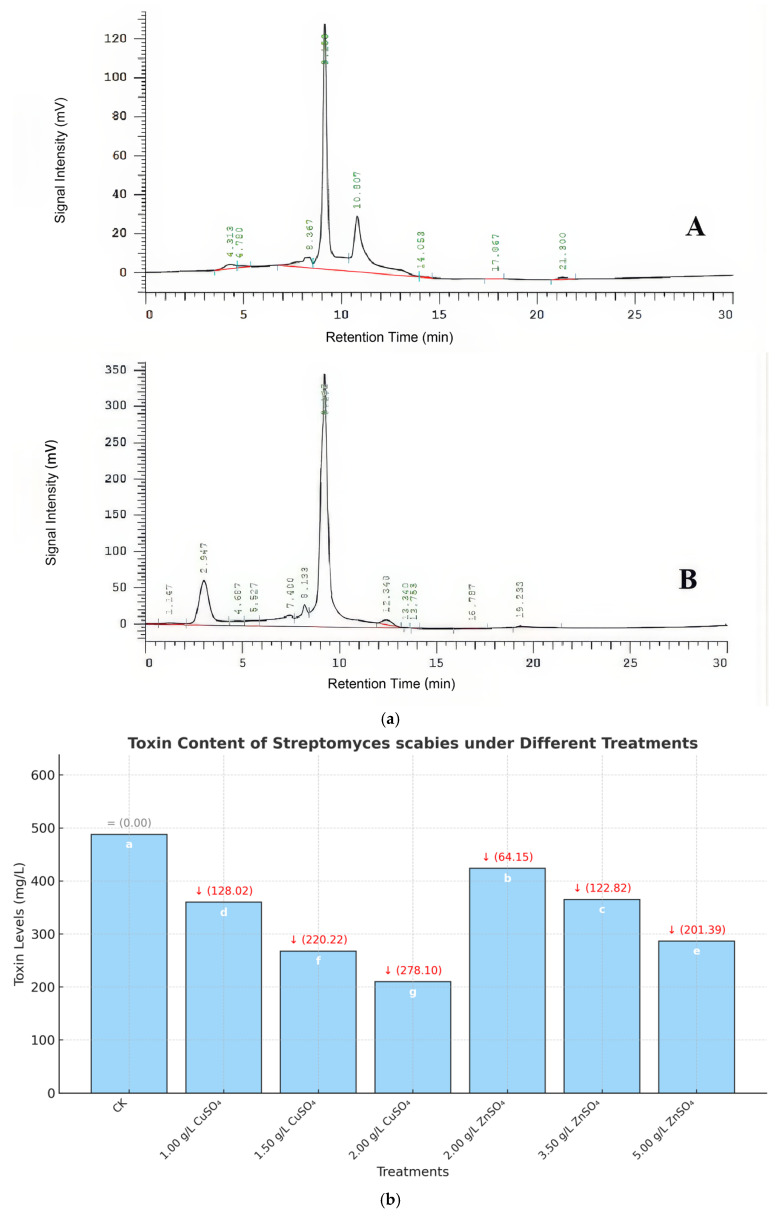
(a) HPLC analysis of Thaxtomin A toxin. Note: A: HPLC for standards, B: HPLC for part of samples. (b) Toxin content produced by *Streptomyces scabies* under different concentrations of medicament treatments. Note: Numbers at the top of each bar represent the reduction in toxin levels (mg/L) compared to the control (CK), which was set as the baseline (0.00 mg/L reduction). Letters inside the bars (a–g) indicate significant differences in toxin levels among treatments, determined by one-way ANOVA followed by Tukey’s post hoc test (*p* < 0.05). For detailed statistics, please refer to Appendix A.

**Figure 5 microorganisms-13-01288-f005:**
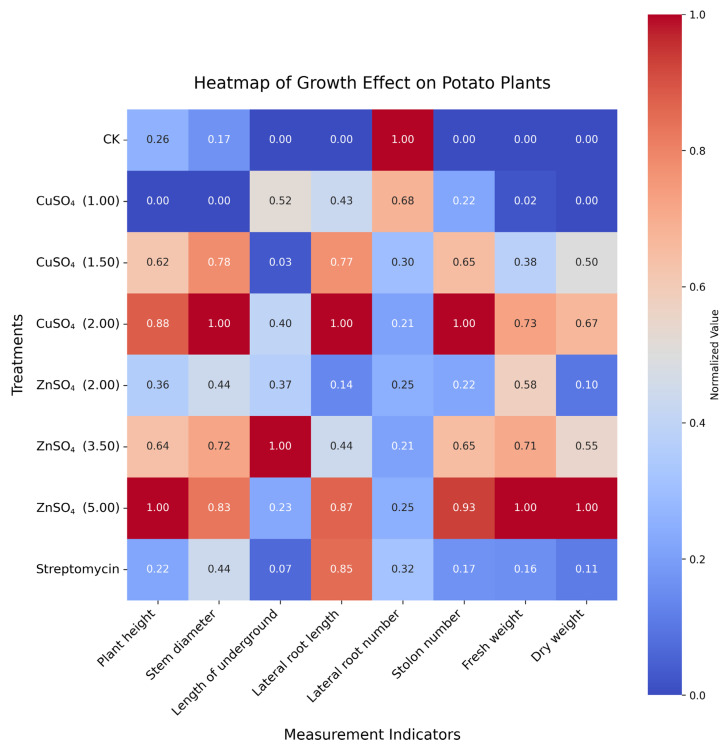
Heatmap of the growth effects of copper sulfate (CuSO_4_) and zinc sulfate (ZnSO_4_) on potato plants under different treatments. Note: Treatments (Y-axis) include the control group (CK); CuSO_4_ at 0.001 g/mL, 0.0015 g/mL, and 0.002 g/mL; ZnSO_4_ at 0.002 g/mL, 0.0035 g/mL, and 0.005 g/mL; and streptomycin at 0.00035 g/mL as a positive control. Measurement indicators (X-axis) include plant height, stem diameter, length of underground (total root length below soil), lateral root length, lateral root number, stolon number, fresh weight, and dry weight. Colors represent normalized values (0.0 to 1.0), with blue (0.0) indicating no improvement and red (1.0) indicating maximum improvement relative to the control group (CK). Significant differences among treatments for each parameter (*p* < 0.05) are detailed in Appendix A.

**Figure 6 microorganisms-13-01288-f006:**
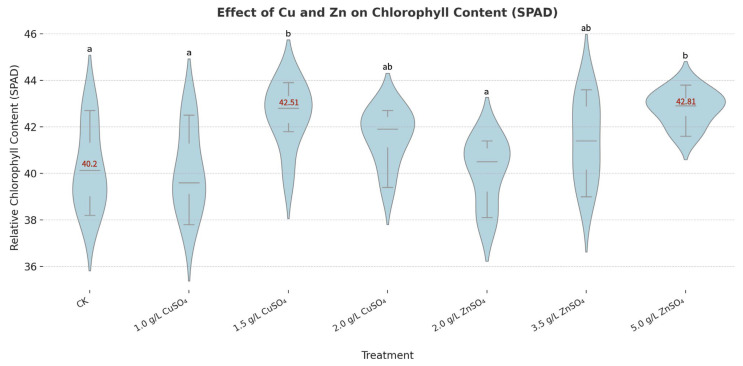
Effects of copper sulfate (CuSO_4_) and zinc sulfate (ZnSO_4_) on chlorophyll content. Note: Letters above the violin plots (a, b, ab) indicate significant differences among treatments, determined by one-way ANOVA followed by Tukey’s post hoc test (*p* < 0.05).

**Figure 7 microorganisms-13-01288-f007:**
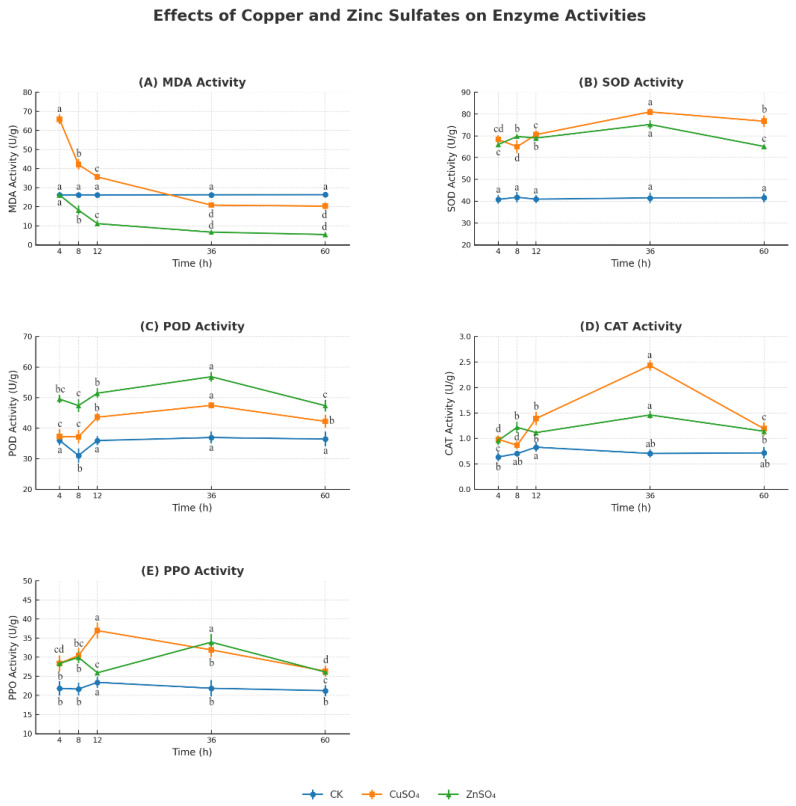
Changes in the activities of SOD, POD, CAT, PPO, and MDA in potato leaves under treatments with 0.002 g/mL CuSO_4_, 0.005 g/mL ZnSO_4_, and control (CK). Data points represent mean values ± standard error (SE) from three biological replicates with three technical replicates each. Significant differences among treatments at each time point (4, 8, 12, 36, and 60 h) are indicated by different lowercase letters (a, b, c, d) (*p* < 0.05), as determined by one-way ANOVA followed by Tukey’s post hoc test.

**Figure 8 microorganisms-13-01288-f008:**
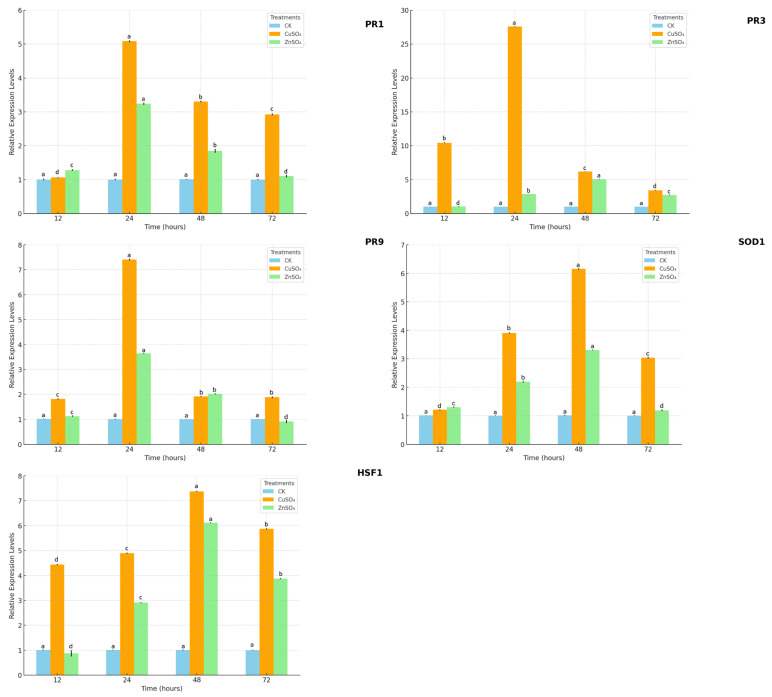
Expression changes of PR1, PR3, PR9, SOD1, and HSF1 genes in potato leaves under treatments with 0.002 g/mL CuSO_4_, 0.005 g/mL ZnSO_4_, and control (CK). Data points represent mean fold changes ± standard error (SE) from three biological replicates with three technical replicates each, calculated using the 2^−ΔΔCt^ method. Significant differences among time points (4, 8, 12, 36, and 60 h) within each treatment and among treatments at each time point were determined by one-way ANOVA followed by Tukey’s post hoc test (*p* < 0.05). Different lowercase letters (a, b, c, d) indicate significant differences among treatments at each time point (*p* < 0.05). Complete statistical results for all genes and time points are available from the corresponding author upon reasonable request.

## Data Availability

The original contributions presented in this study are included in the article/Appendix A. Further inquiries can be directed to the corresponding author.

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
