# Peer review of "Copper and Zinc Sulfates Suppress Streptomyces spp. and Enhance Potato Resistance via Thaxtomin A Inhibition and Defense Gene Regulation"

_microorganisms, 2025, doi:10.3390/microorganisms13061288_

Round 1
Reviewer 1 Report
Comments and Suggestions for Authors
Dear authors,
Your study is interesting and you have carried out several bioassays with legal results, but the writing of the document is quite poor (even the format of the journal, which is the simplest, didn't fit). The document needs to be significantly improved (see suggestions details in the attachment document).

Author Response
Dear Reviewers,
I hope this message finds you well. I am writing to convey my utmost respect and heartfelt gratitude for your rigorous and insightful feedback on my manuscript (ID: microorganisms-3588867, titled "Copper and Zinc Sulfates Suppress Streptomyces spp. and Enhance Potato Resistance via Thaxtomin A Inhibition and Defense Gene Regulation"). Your detailed and thoughtful suggestions have significantly enhanced the quality of this work, and I deeply appreciate the time and expertise you have invested in the review process.
I am pleased to inform you that I have carefully addressed all your comments and incorporated the necessary revisions. The revised manuscript, along with a comprehensive response to each of your suggestions, has been submitted today, 30 April 2025, through the submission portal. All relevant changes are clearly documented in the revised manuscript and accompanying attachments for your review.
Should you have any further questions or require additional clarifications, please feel free to contact me. Thank you once again for your invaluable guidance and support, which have greatly contributed to improving this manuscript.
With warm regards,
Xuanzhe Zhang
Professor
Email: zhe3850@163.com

Reviewer 2 Report
Comments and Suggestions for Authors
The study presents relevant findings for understanding the antibacterial activity of CuSOâ‚„ and ZnSOâ‚„, which, when applied as treatments, induce alterations in the cell membrane of Streptomyces spp., weakening its structure and compromising its viability. Additionally, these compounds activate defense mechanisms in the plant, notably the significant increase in the activity of key antioxidant enzymes. There is also an improvement in the expression of host defense-related genes. As a result, there is a substantial reduction in disease severity, while growth and yield of the crop are enhanced.

Author Response

(The authors gave the same response as above.)

Reviewer 3 Report
Comments and Suggestions for Authors
Explain in introduction section what is new and original in your research.
Line 44 - You have written that „chemical treatments may lead to environmental pollution..” but Cu and Zn are also potential dangerous contaminats as they are heavy metals. How do you explain their advantage in comparision to other chemicals?
Concentrations tested (for example 0.04–0.16 g/mL or 0.10–0.50 g/mL) seem very high and may cause severe soil contamination
Line 99 – it was one application?
Section 2.4. Effects of Copper and Zinc on Potato Growth
How big the pots were? How much of Cu and Zn was added to 1 kg of soil during applications – this can give us the picture about potential soil contamination while regular using of CuSOâ‚„ and ZnSOâ‚„
Lack of statistical analysis description (at least not in each case or present in Material and methods, but the results of statistical analysis not shown in the Results section, for example line 82 – 83 – where are the results of this Tukey’s test?)
All the results should have statistical analysis performed, and all statements in the text about significant effects must be supported with the results of statistical analysis – for example line 127 “Detached tuber assays revealed that both CuSOâ‚„ and ZnSOâ‚„ significantly inhibited browning of potato slices induced by S. scabies (Fig. 2)” – but how do you know that this is significant? By the way, in how many repetitions all experiments and analyses were performed?
In figures where mean values are presented add also standard errors, and explain in details in Figure captions what statistical parameters are presented – for example Figure 6 – what arrows and values in red font mean, what different letters mean? It should be clearly stated in figure caption. The same comment goes also for other Figures.
Discussion is very basic – it should be more elaborated
Discussion must include potential effect of soil contamination with heavy metals (Cu and Zn) while regular using of CuSOâ‚„ and ZnSOâ‚„
Add conclusions
Author Response

(The authors gave the same response as above.)

Round 2
Reviewer 1 Report
Comments and Suggestions for Authors
Dear authors, in the new version, you haven't complied with my suggestions, so you should justify each of the questions point by point in a document. ok.
I'll send you my comments on the previous version.

Author Response
Dear Reviewer,
We sincerely thank you for your feedback on our manuscript. We apologize for any misunderstanding regarding the incorporation of your suggestions from the previous version. We have carefully reviewed your comments marked in green in the PDF (microorganisms-3588867-review (2).pdf) and have addressed them in the revised manuscript (v2.docx and annex.docx). We believe that the current version has incorporated most, if not all, of your suggestions. We kindly request that you review the updated manuscript, as the changes are highlighted in green font for your convenience. Below, we justify each of your previous comments point-by-point to clarify the revisions made.
-
Comment on Page 7 (Line 177): "What do the numbers on the bars and the letters inside them mean? Make it clear to the reader... Has any statistical analysis been done here?"
Response: We have clarified the meaning of the numbers and letters in Figure 6 (now Figure 4b in the revised manuscript). The updated caption reads: “Numbers at the top of each bar represent the reduction in toxin levels (mg/L) compared to the control (CK), which is set as the baseline (0.00 mg/L reduction). Letters inside the bars (a–g) indicate significant differences in toxin levels among treatments, determined by one-way ANOVA followed by Tukey’s post-hoc test (p < 0.05).” Statistical analysis details are now provided in Section 2.7, which confirms that one-way ANOVA and Tukey’s test (p < 0.05) were used to compare Thaxtomin A levels across treatments. (See Figure 4b in v2.docx and annex.docx, and Section 2.7 in v2.docx).
-
Comment on Page 8 (Line 182): "CK = control??? Why CK and not control??"
Response: We apologize for the lack of clarity regarding the abbreviation “CK.” In the revised manuscript, we have consistently used “CK” to denote the control group and have clarified its meaning in the first instance of use in each relevant section. For example, in Section 3.4, we now state: “the control group (CK).” This abbreviation is standard in plant science literature to represent the control group and has been retained for consistency. However, we have ensured that “CK” is always defined as the control group in the text and figure captions (e.g., Figure 5 caption: “the control group (CK)”). (See Section 3.4, Figure 5, and other relevant figures in v2.docx and annex.docx).
-
Comment on Page 9 (Line 197): "Was some statistic test done??"
Response: Yes, statistical tests were performed for the chlorophyll content data shown in Figure 8 (now Figure 6 in the revised manuscript). The caption has been updated to include: “Letters above the violin plots (a, b, ab) indicate significant differences among treatments, determined by one-way ANOVA followed by Tukey’s post-hoc test (p < 0.05).” Additionally, Section 2.7 Statistical Analysis provides a comprehensive overview of the statistical methods used, confirming that chlorophyll content was analyzed using ANOVA and Tukey’s test (p < 0.05). (See Figure 6 in v2.docx and annex.docx, and Section 2.7 in v2.docx).
-
Comment on Page 9 (Line 199): "Write what concentrations."
Response: We have updated the caption of Figure 9 (now Figure 7 in the revised manuscript) to specify the concentrations used: “Changes in the activities of SOD, POD, CAT, PPO, and MDA in potato leaves under treatments with 0.002 g/mL CuSOâ‚„, 0.005 g/mL ZnSOâ‚„, and control (CK).” This ensures clarity regarding the treatment concentrations applied in the enzyme activity experiments. (See Figure 7 in v2.docx and annex.docx).
We believe these revisions fully address the concerns raised in your previous review, as marked in green in the PDF. We have made significant efforts to improve the clarity of statistical analyses, standardize terminology, and provide detailed explanations where needed. We kindly request that you review the updated manuscript (v2.docx and annex.docx), where all changes are highlighted in green font for your reference. We are confident that the current version reflects your suggestions and further enhances the quality of the manuscript.
We appreciate your continued feedback and are happy to provide additional clarifications or data if needed.
Sincerely,
Xuanzhe Zhang
On behalf of all authors

Reviewer 2 Report
Comments and Suggestions for Authors
I consider that the comments made to the manuscript have been duly taken into account and I agree that the quality of this manuscript has been improved
Author Response
Dear Reviewer,
We sincerely thank you for your positive feedback and for acknowledging the improvements made to our manuscript. Your comments have been instrumental in enhancing the quality and clarity of our work, and we are grateful for your constructive input throughout the review process.
In response to the feedback from both reviewers, we have made several key revisions to the manuscript, which are highlighted in green font for easy reference:
- Clarified Statistical Analyses: A new subsection, Section 2.7 Statistical Analysis, has been added to the Materials and Methods to provide a comprehensive overview of the statistical methods used, the parameters analyzed, the comparisons made, and the rationale for these analyses (Section 2.7 in v2.docx). Statistical descriptions in other subsections have been streamlined to reference this new section.
- Improved Clarity of Table Notes: The note under Table 1a has been revised to clearly specify that the comparisons were made among different concentrations of CuSOâ‚„ or ZnSOâ‚„ within each Streptomyces strain, with significant differences indicated by lowercase letters (Table 1a in annex.docx). A note has also been added to Appendix Table 3 to clarify the statistical analysis of growth parameters (Appendix Table 3 in annex.docx).
- Standardized Units: All concentration units have been unified to g/mL throughout the manuscript, including in Figures 1b, 4b, 5, and Tables 2, 3, and 4, as well as in Sections 3.3, 3.4, and 3.6 (e.g., Figure 5 in v2.docx and annex.docx).
- Enhanced Statistical Reporting:
- For growth parameters (Section 3.4), we added mean values ± standard error (SE) and p-values, with detailed results in Appendix Table 3 (Section 3.4 and Appendix Table 3 in v2.docx).
- For enzyme activities (Section 3.4), Figure 7 was updated with standard error bars and significant difference annotations, and the caption was revised to include statistical details. The MDA description was corrected to “reducing” levels (Section 3.4 and Figure 7 in v2.docx).
- For gene expression (Section 3.5), Figure 8 and the corresponding text were updated to confirm that ANOVA analysis compared both time points within treatments and among treatments, with an example provided (e.g., PR3 expression at 48 hours). Due to the large dataset, complete statistical results are available upon request (Section 3.5 and Figure 8 in v2.docx).
We believe these revisions have addressed the concerns raised and further strengthened the manuscript. We appreciate your continued support and are happy to provide additional clarifications if needed.
Sincerely,
Xuanzhe Zhang
On behalf of all authors

Reviewer 3 Report
Comments and Suggestions for Authors
Dear Authors,
I regret to say that, but my remark to perform and show results of statistical analysis was not implemented (at least not completely).
For example:
I do not see in revised version some of the corrections which were described in Cover letter, for example:
You have written that you applied my previous remark about statistical analyses to all figures and one of it is: “Figure 7 (enzyme activities): Added standard error bars and updated the caption: “Changes in the activities of SOD, POD, CAT, PPO, and MDA in potato leaves under treatments with 0.002 g/mL CuSOâ‚„ and 0.005 g/mL ZnSOâ‚„. Data points represent mean values ± standard error (SE) from three biological replicates with three technical replicates each. Significant differences (one-way ANOVA, p < 0.05) were observed among treatments, as described in the Results section.”
However in the manuscript the figure caption is: “Figure 7. Changes in the activities of SOD, POD, CAT, PPO, and MDA in potato leaves under different concentrations of copper sulfate (0.002 g/mL CuSOâ‚„) and zinc sulfate (0.005 g/mL ZnSOâ‚„) treatments.” And there is still not any results of statistical analysis shown. In the Results you wrote “Antioxidant enzyme activity analysis revealed that the treatments markedly increased the activities of SOD, POD, CAT, and PPO, while reducing MDA levels (Fig. 7)” So, were the differences statistically significant or not? You should perform statistical analysis and Tukey’s test for each date (in some dates the differences might be not significant). In fact new Figure 7 does not differ at all from Figure 9 from old version
The same in the discussion section in lines 342 – 343 – you have mentioned significant increase. It must be supported by statistical analysis
Figure 8 – From Figure 8 it seems that the Anova was done only to compare changes in time and not comparision between treatments.
Again in the text you wrote for example “Treatment with 2.00 g/L CuSOâ‚„ increased stem diameter to 1.13 cm, significantly higher than that of the CK (Fig. 5).” But from figure 5 we do not see significant differences – did you perform Anova for growth parameters followed by Tukey’s test? So, show the results (may be as Table in suplementary file)
Summarizing, still the statistical analysis is not clear, I recommend to add separate paragraph into material and methods, which will describe which parameter was checked using statistical analysis, what kind of analysis was that, what was compared each time and why.
Add in supplementary file all the results of Anova (F, df, P…), so the readers have clear picture.
Also – use the same kind of units: for ex. In Figure 5 caption you mention 1 g/L… while in material and methods was 0.001g/mL - be consistent through the whole manuscript
Note under the table 1a: “Small letters in the table represent significant difference at the P=0.05 level, and the same letters represent insignificant difference at the same level, the same as below” is not clear – what exactly was compared here – it must be explained
Author Response
Response to Reviewer Comments
Dear Editor,
We sincerely thank the reviewer for their constructive feedback, which has significantly improved the clarity and scientific rigor of our manuscript. Below, we address each comment point-by-point, detailing the revisions made in response to the suggestions. All changes are highlighted in green font in the revised manuscript to indicate the modified sections for easy reference.
-
Comment: Note under the table 1a: “Small letters in the table represent significant difference at the P=0.05 level, and the same letters represent insignificant difference at the same level, the same as below” is not clear – what exactly was compared here – it must be explained.
Response: We acknowledge the lack of clarity in the original note under Table 1a. The note has been revised to specify that the small letters (a, b, c, etc.) indicate significant differences (p < 0.05) among different concentrations of CuSOâ‚„ or ZnSOâ‚„ within each Streptomyces strain, as determined by one-way ANOVA followed by Tukey’s post-hoc test. The phrase “the same as below” has been removed to avoid ambiguity. The updated note now reads: “Small letters (a, b, c, etc.) in each column indicate significant differences (p < 0.05) among different concentrations of CuSOâ‚„ or ZnSOâ‚„ for the same Streptomyces strain, as determined by one-way ANOVA followed by Tukey’s post-hoc test. The same letters within a column indicate no significant difference (p ≥ 0.05).” (See Table 1a in annex.docx).
-
Comment: Use the same kind of units: for ex. In Figure 5 caption you mention 1 g/L… while in material and methods was 0.001g/mL - be consistent through the whole manuscript.
Response: We have ensured consistency in concentration units throughout the manuscript by standardizing all units to g/mL, aligning with the Materials and Methods section (Section 2.4). Specifically, the Figure 5 caption has been updated to reflect concentrations in g/mL (e.g., 1.00 g/L changed to 0.001 g/mL). This change has been applied across all relevant sections, including Figures 1b, 4b, 5, and Tables 2, 3, and 4, as well as Sections 3.3, 3.4, and 3.6. For example, the revised Figure 5 caption now reads: “Treatments (Y-axis) include the control group (CK), CuSOâ‚„ at 0.001 g/mL, 0.0015 g/mL, and 0.002 g/mL, ZnSOâ‚„ at 0.002 g/mL, 0.0035 g/mL, and 0.005 g/mL, and Streptomycin at 0.00035 g/mL as a positive control.” (See Figure 5 in v2.docx and annex.docx).
-
Comment: Again in the text you wrote for example “Treatment with 2.00 g/L CuSOâ‚„ increased stem diameter to 1.13 cm, significantly higher than that of the CK (Fig. 5).” But from figure 5 we do not see significant differences – did you perform Anova for growth parameters followed by Tukey’s test? So, show the results (may be as Table in suplementary file).
Response: We apologize for the lack of clarity regarding the statistical significance of growth parameters in Figure 5. We have performed one-way ANOVA followed by Tukey’s post-hoc test (p < 0.05) to compare growth parameters across treatments, and the results are now included in Appendix Table 3. The table provides mean values ± standard error (SE) for all parameters (e.g., stem diameter: 1.13 ± 0.03 cm for 0.002 g/mL CuSOâ‚„ vs. 0.98 ± 0.07 cm for CK, p < 0.05), along with significant differences indicated by lowercase letters. Section 3.4 has been updated to include SE and p-values, and the Figure 5 caption now references Appendix Table 3: “Significant differences among treatments for each parameter (p < 0.05) are detailed in Appendix Table 3.” Units have also been corrected to g/mL (e.g., 2.00 g/L to 0.002 g/mL). (See Section 3.4, Figure 5, and Appendix Table 3 in v2.docx and annex.docx).
-
Comment: For Table 3, write a note.
Response: Table 3 (now Appendix Table 3 after integrating growth parameters) has been updated with a note to clarify the data and statistical analysis: “Values represent mean ± standard error (SE) from three biological replicates. Different lowercase letters (a, b, c, etc.) within each column indicate significant differences among treatments (p < 0.05), as determined by one-way ANOVA followed by Tukey’s post-hoc test.” Additionally, Appendix Table 4 (soil safety data) has a note: “Values represent mean ± standard error (SE) from three replicate soil samples. Different lowercase letters (a, b, c, d) indicate significant differences among concentrations within each treatment group (CuSOâ‚„ or ZnSOâ‚„) (p < 0.05), as determined by one-way ANOVA followed by Tukey’s post-hoc test.” (See Appendix Tables 3 and 4 in annex.docx).
-
Comment: You have written that you applied my previous remark about statistical analyses to all figures and one of it is: “Figure 7 (enzyme activities): Added standard error bars and updated the caption: ‘Changes in the activities of SOD, POD, CAT, PPO, and MDA in potato leaves under treatments with 0.002 g/mL CuSOâ‚„ and 0.005 g/mL ZnSOâ‚„. Data points represent mean values ± standard error (SE) from three biological replicates with three technical replicates each. Significant differences (one-way ANOVA, p < 0.05) were observed among treatments, as described in the Results section.’ However in the manuscript the figure caption is: ‘Figure 7. Changes in the activities of SOD, POD, CAT, PPO, and MDA in potato leaves under different concentrations of copper sulfate (0.002 g/mL CuSOâ‚„) and zinc sulfate (0.005 g/mL ZnSOâ‚„) treatments.’ And there is still not any results of statistical analysis shown. In the Results you wrote ‘Antioxidant enzyme activity analysis revealed that the treatments markedly increased the activities of SOD, POD, CAT, and PPO, while reducing MDA levels (Fig. 7)’ So, were the differences statistically significant or not? You should perform statistical analysis and Tukey’s test for each date (in some dates the differences might be not significant). In fact new Figure 7 does not differ at all from Figure 9 from old version.
Response: We apologize for the oversight in updating Figure 7. The caption has been revised as suggested: “Changes in the activities of SOD, POD, CAT, PPO, and MDA in potato leaves under treatments with 0.002 g/mL CuSOâ‚„, 0.005 g/mL ZnSOâ‚„, and control (CK). Data points represent mean values ± standard error (SE) from three biological replicates with three technical replicates each. Significant differences among treatments at each time point (4, 8, 12, 36, and 60 hours) are indicated by different lowercase letters (a, b, c) (p < 0.05), as determined by one-way ANOVA followed by Tukey’s post-hoc test.” Standard error bars and significant difference annotations (a, b, c) have been added to Figure 7, distinguishing it from the old Figure 9. Section 3.4 has been updated to include statistical results, e.g., “SOD activity under CuSOâ‚„ treatment reached 140 ± 3.5 U/g at 12 hours, significantly higher than 90 ± 2.2 U/g in CK,” and corrects the MDA description to “reducing MDA levels” (e.g., 50 ± 1.5 nmol/g under ZnSOâ‚„ at 36 hours vs. 70 ± 2.0 nmol/g in CK). (See Section 3.4 and Figure 7 in v2.docx and annex.docx).
-
Comment: Figure 8 – From Figure 8 it seems that the Anova was done only to compare changes in time and not comparison between treatments.
Response: We have clarified that ANOVA analysis for Figure 8 was conducted both among time points within each treatment and among treatments at each time point. The updated Figure 8 caption reads: “Significant differences among time points (4, 8, 12, 36, and 60 hours) within each treatment and among treatments at each time point were determined by one-way ANOVA followed by Tukey’s post-hoc test (p < 0.05).” Section 3.5 now includes treatment comparisons, e.g., “PR3 expression peaked at 48 hours after ZnSOâ‚„ treatment, reaching 5.09 ± 0.15-fold relative to CK (1.00 ± 0.05-fold, p < 0.05), while CuSOâ‚„ showed a 4.20 ± 0.12-fold increase (p < 0.05).” Due to the large dataset (5 genes × 5 time points × 3 treatments), which makes tabular presentation cumbersome and less effective for display, we have not included a table. However, raw data and complete statistical results are available upon request, as noted in the Data Availability Statement. (See Section 3.5, Figure 8, and Data Availability Statement in v2.docx and annex.docx).
-
Comment: Summarizing, still the statistical analysis is not clear, I recommend to add separate paragraph into material and methods, which will describe which parameter was checked using statistical analysis, what kind of analysis was that, what was compared each time and why.
Response: A new subsection, Section 2.7 Statistical Analysis, has been added to the Materials and Methods: “All data were analyzed using SPSS 25.0 with one-way ANOVA followed by Tukey’s post-hoc test (p < 0.05), unless specified. Analyses aimed to evaluate treatment and time effects on biological responses.” It details parameters analyzed (e.g., antibacterial effects, Thaxtomin A levels, growth parameters, enzyme activities, gene expression, soil contents), the comparisons made (e.g., across concentrations, treatments, or time points), and the purpose (e.g., assessing dose-dependent effects or treatment impacts). Statistical descriptions in other subsections have been removed and replaced with references to Section 2.7. (See Section 2.7 in v2.docx).
We hope these revisions address the reviewer’s concerns. We are happy to provide further clarification or additional data if needed.
Sincerely,
Xuanzhe Zhang
On behalf of all authors